

# A comprehensive dataset on biomechanics and motor control during human walking with discrete mechanical perturbations

Dana L. Lorenz[1] and Antonie J. van den Bogert[2]

[1] Department of Chemical and Biomedical Engineering, Cleveland State University, Cleveland, Ohio, United States
[2] Department of Mechanical Engineering, Cleveland State University, Cleveland, Ohio, United States

Corresponding author
Dana L. Lorenz,
d.l.lorenz@vikes.csuohio.edu

## ABSTRACT

**Background:** Humans have a remarkable capability to maintain balance while walking. There is, however, a lack of publicly available research data on reactive responses to destabilizing perturbations during gait.

**Methods:** Here, we share a comprehensive dataset collected from 10 participants who experienced random perturbations while walking on an instrumented treadmill. Each participant performed six 5-min walking trials at a rate of 1.2 m/s, during which rapid belt speed perturbations could occur during the participant's stance phase. Each gait cycle had a 17% probability of being perturbed. The perturbations consisted of an increase of belt speed by 0.75 m/s, delivered with equal probability at 10%, 20%, 30%, 40%, 50%, 60%, 70%, or 80% of the stance phase. Data were recorded using motion capture with 25 markers, eight inertial measurement units (IMUs), and electromyography (EMG) from the tibialis anterior (TA), soleus (SOL), lateral gastrocnemius (LG), rectus femoris (RF), vastus lateralis (VL), vastus medialis (VM), biceps femoris (BF), and gluteus maximus (GM). The full protocol is described in detail.

**Results:** We provide marker trajectories, force plate data, EMG data, and belt speed information for all trials and participants. IMU data is provided for most participants. This data can be useful for identifying neural feedback control in human gait, biologically inspired control systems for robots, and the development of clinical applications.

## INTRODUCTION

Human walking is inherently unstable, and the feedback control mechanisms to maintain balance are complex, including spinal reflexes as well as vestibular and visual feedback. Reflexes play an essential role in ensuring that corrective actions are taken quickly after an unexpected perturbation due to uneven ground or external forces (*Nielsen & Sinkjaer, 2002*). Absence or insufficiency of reflexes would increase the risk of falling and injury. On the other hand, hyperactive reflexes can cause abnormal movements with certain

neurological disorders, such as cerebral palsy and stroke (*Flux et al., 2021*; *Li & Francisco, 2015*). Clinically, reflexes can be evaluated with the familiar tendon tap tests. During gait, the Hoffmann reflex (H-reflex) can be used, which involves electrical stimulation of a sensory nerve, which excites the stretch reflex pathway and causes a muscle activation response that can be detected with electromyography (EMG) (*Belanger & Patla, 1984*; *Zehr, Komiyama & Stein, 1997*; *Zehr, Stein & Komiyama, 1998*).

However, reactive responses to perturbations require mechanisms beyond reflexes. Strategies to recover from perturbations can involve longer time scales and complex neural mechanisms, such as altered foot placement in the contralateral limb (*Bruijn & van Dieën, 2018*). Electrical stimulation or clinical reflex tests will provide limited insight into the capability of a patient or research participant to maintain balance, and the neural control mechanisms that are responsible.

A more ecological approach is the use of mechanical perturbations to elicit reactive responses during a functional activity such as gait. Wearable exoskeletons have been used to induce stretch reflexes during walking (*Stevenson et al., 2015*), which is suitable for basic research, but the weight and discomfort of this approach are important limitations. Recently, it has been demonstrated that rapid changes in belt speed can induce reflexes during treadmill walking (*Sloot et al., 2015*). Such perturbations are similar to those that occur naturally but with controlled timing and magnitude. In *Sloot et al.*'s *(2015)* work, perturbations were applied only at one time in the gait cycle, but it is known from other studies that reflex gains are modulated throughout the gait cycle (*Zehr & Stein, 1999*). To better understand reflexes and the mechanisms behind their control we need to look at responses to perturbations that occur at multiple points in the gait cycle. It is also important to look at the entire perturbation response, including non-reflex responses on a longer time scale, and using full body biomechanical measures in addition to electromyography.

In recent years, it has been emphasized that open datasets are of vital importance to aid scientific advancements by inspiring new analyses, reproducibility, and good data practices (*McKiernan et al., 2016*). One of the most widely used datasets for human gait is a set of normative data from David Winter (*Winter, 2009*). It is only for a small number of participants over a few gait cycles, yet it has enabled many studies, like control design for prosthetics (*Sup, Bohara & Goldfarb, 2008*). More extensive datasets, including gait during various speeds and conditions, have since been published (*Fukuchi, Fukuchi & Duarte, 2018*), often including motion capture, ground reaction forces (GRF), and EMG. Datasets that include data from inertial measurement units (IMUs), together with conventional instrumentation, are especially valuable because this allows the validation of IMU processing methods (*Camargo et al., 2021*). However, there are limited publicly available data on human gait during mechanical perturbations.

*Moore, Hnat & van den Bogert (2015)* published data from fifteen participants walking at three different speeds, during normal walking and during walking with pseudo-random belt speed fluctuations. However, the pseudo-random perturbations, though destabilizing, were small, the standard deviation of belt speed variations was 0.06–0.21 m/s, and may not

have exceeded the sensory threshold for reflex responses (*Moore, Hnat & van den Bogert, 2015*). *Liu, Park & Finley (2022)* shared data from eleven participants walking at their preferred walking speed with six different perturbation magnitudes with the objective of determining if the reference point influences how dynamic balance is controlled during perturbed walking. *Leestma et al. (2023)* shared data from eleven participants walking with perturbations caused by platform movement in the transverse plane. However, these datasets do not include EMG data, which can provide information about neuromuscular mechanisms (*Huang et al., 2011*).

This article presents able-bodied human walking data from 10 participants. The participants walked on a treadmill while belt speed changes were generated at controlled times during randomly selected gait cycles. Motion, ground reaction force, and EMG data were recorded. Motion capture was performed with IMUs in addition to cameras. The data has been organized and made available following best data-sharing practices for other researchers (*White et al., 2013*). We expect that this work will assist data-driven analyses for the development of models and control systems for robotics and lay the groundwork for clinical applications of neural control identification.

## METHODS

### Participants

For this study, ten able-bodied participants, five male and five female, participated with an average age of 21.60 ± 4.06 years, height of 1.69 ± 0.11 m, and mass of 63.70 ± 9.44 kg. The study was approved by the Institutional Review Board of Cleveland State University (IRB-FY2019-178). Before the experiment began written, informed consent was attained from participants.

To protect the identity of the participants each participant was assigned a unique participant ID number. The general information gathered for each participant is shown in Table 1.

### Equipment

The data were collected in the Human Motion and Control Laboratory at Cleveland State University. The instrumentation as described by *Moore, Hnat & van den Bogert (2015)* includes and R-Mill instrumented treadmill (Forcelink, Culemborg, Netherlands) and motion capture system. Additionally, Trigno Legacy EMG Sensors (Delsys Inc., Natick, MA, USA) with analog EMG and three channel Dual Range Accelerometer were used to gather analog EMG data and Trigno IM Sensors (Delsys Inc., Natick, MA, USA) with digital EMG and nine channel Quad range IMU chip were used to gather inertial measurement unit data.

Similarly to *Moore, Hnat & van den Bogert (2015)*, Cortex 5.5.01 software (Motion Analysis, Santa Rosa, CA, USA) received the force plate data, analog sensor (EMG) data, and 3D marker data *via* a National Instruments USB-6255 data acquisition unit. Cortex sends the data, in real-time to the D-Flow 3.34.0 software (Motek Medical, Amsterdam, Netherlands), which is responsible for controlling the speed of the treadmill belts. The

**Table 1 Information about the 10 participants.**

| ID | Sex | Age (years) | Height (m) | Mass (kg.) |
|----|-----|-------------|------------|------------|
| 02 | Female | 25 | 1.65 | 68.03 |
| 03 | Male | 18 | 1.83 | 60.33 |
| 04 | Female | 23 | 1.57 | 56.25 |
| 15 | Female | 30 | 1.59 | 59.14 |
| 51 | Male | 18 | 1.83 | 64.41 |
| 66 | Female | 25 | 1.70 | 69.40 |
| 76 | Male | 19 | 1.75 | 58.97 |
| 87 | Female | 21 | 1.49 | 49.90 |
| 90 | Male | 19 | 1.70 | 84.82 |
| 99 | Male | 18 | 1.71 | 65.77 |

**Note:**
   Each participant's sex, age, height, and mass.

force plate and marker data were recorded at a rate of 100 Hz, while the analog EMG data was recorded at 1,000 Hz.

A Delsys Trigger Module (Delsys Inc., Natick, MA, USA) and Phidget InterfaceKit 8/8/8 1018_2 (Phidgets Inc., Calgary, Canada) were used to synchronize the Trigno IMU sensors with the existing laboratory setup. The trigger module was plugged into the IM sensor base and connected to the Phidget board; which was connected to the D-flow computer *via* a USB connection. When recording began in the D-Flow program a 5 V signal was sent to the trigger module, beginning recording of IMU sensor data in the EMGworks 4.7.9 software (Delsys Inc., Natick, MA, USA).

The layout of the treadmill and motion capture equipment, similar to *Moore, Hnat & van den Bogert (2015)* can be seen in Fig. 1. The walking surface of the treadmill is 500 × 1,800 mm, for each belt. The coordinate system of the motion capture system is defined so that the positive z-direction is pointing towards the back of the treadmill, the positive y-direction points upward, and the positive x-direction is to the right. When calibrating the cameras, a point on the treadmill's surface is set as the origin point for the coordinate system, it was also used as the ground reaction force measuring system origin point (*Moore, Hnat & van den Bogert, 2015*).

A flow chart showing how the instrumentation interacts with the computers can be seen in Fig. 2. The figure also shows the file names that the protocol produces, along with a brief description of what can be found in each file.

## Perturbation signals

A Lua script was created within the 'Script' module of the D-Flow software to generate the perturbations and control the speed of the treadmill belts. The script received vertical ground reaction force data from the Mocap module in D-Flow. Once the treadmill was at the normal walking speed of 1.2 m/s, the vertical GRF was used to keep track of the participant's gait cycles, specifically when they entered and exited the stance phase (heel strike and toe-off, respectively) to determine the average duration of their unperturbed stance phase. At heel strike, a random number generator determines if a perturbation will

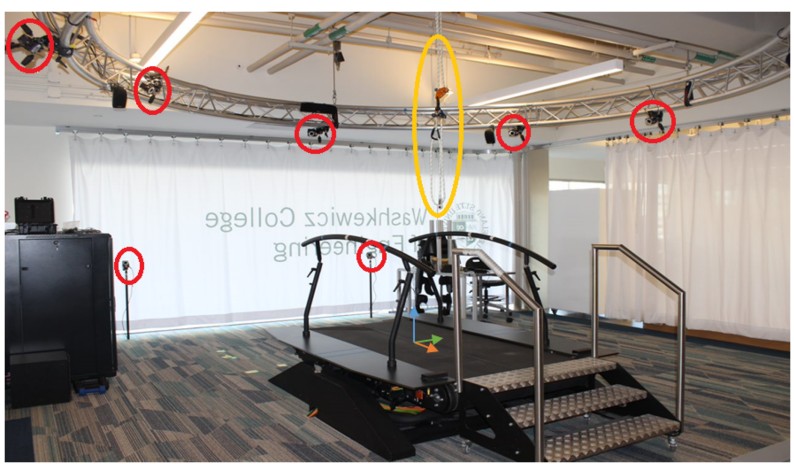

**Figure 1 Laboratory set up of the motion capture and treadmill.** The treadmill system with the x (green), y (blue), and z (orange) coordinate system, cameras (circled in red), and safety rope (circled in yellow). The walking surface of the treadmill is 500 × 1,800 mm for each. The treadmill belt moves in the positive Z direction, with the participant facing the negative Z direction during walking.

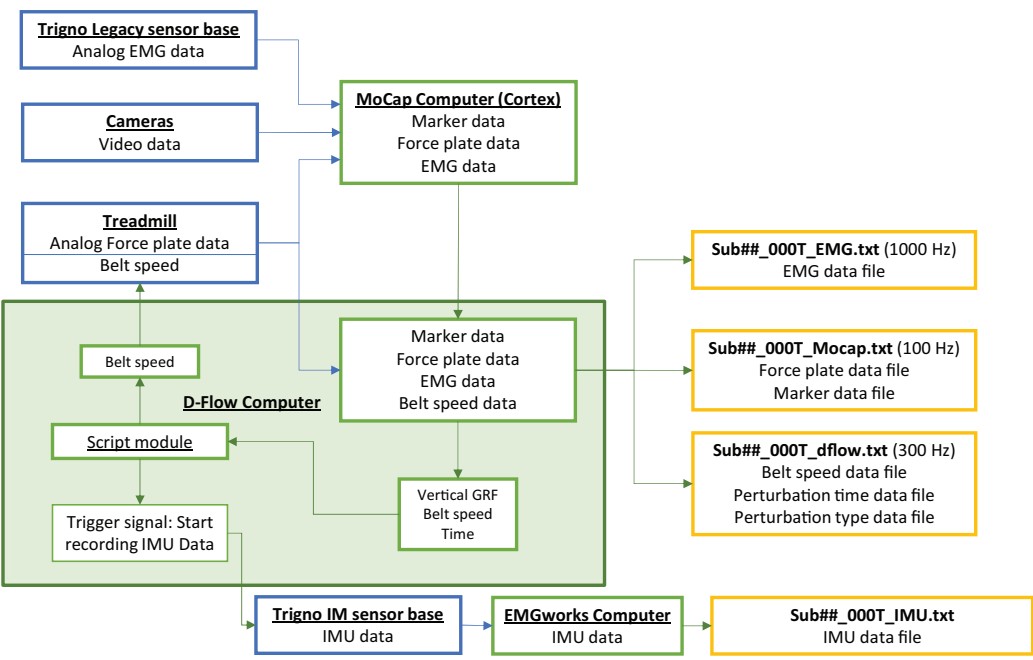

**Figure 2 A flow chart of the experimental setup.** The flow chart shows how data from each hardware component (blue) is shared with the computers (green) and exported to text files (orange) for analysis. In the exported file names '##' indicates the participant number.

occur during that gait cycle. There was a 1 in 6 (about 17%) chance that a perturbation will occur for any gait cycle, except for the two gait cycles immediately following a perturbation. If a perturbation is set to occur, another random number generator determines what type of perturbation (10%, 20%, … or 80%) will occur during that gait cycle. A running average of the stance duration was used determine when the command

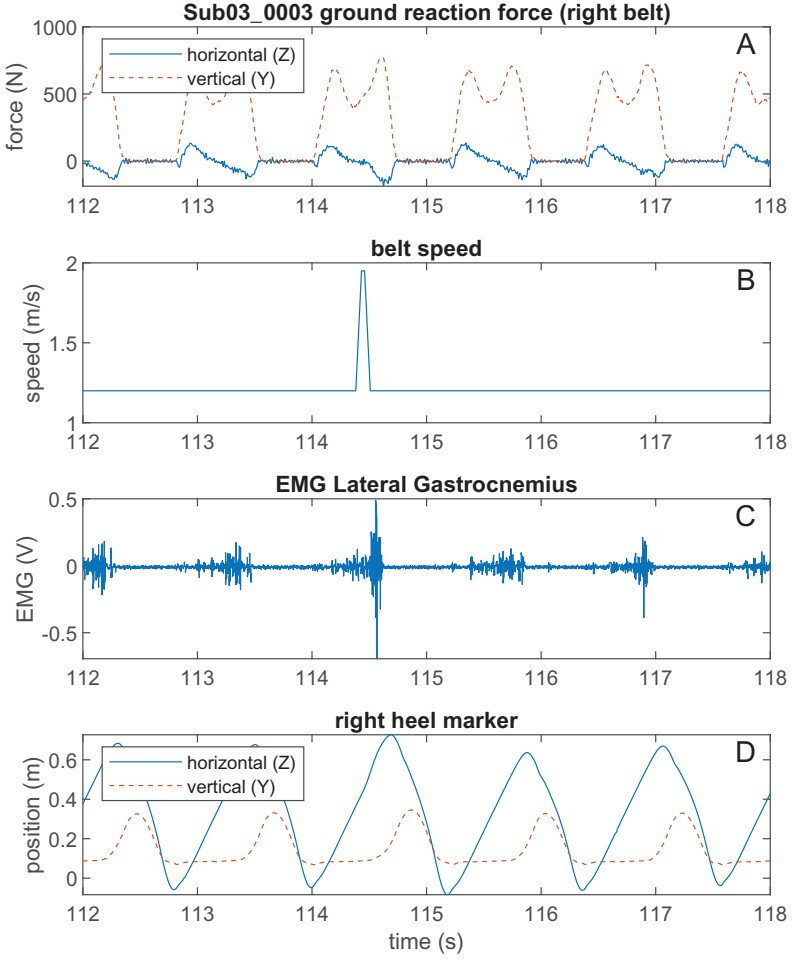

**Figure 3 Example of raw data during a perturbation.** (A) The vertical and horizontal GRF of Participant 3's right side. (B) The speed of the right treadmill belt, where a perturbation can be seen occurring at mid-late stance. (C) The EMG signal of Participant 3's lateral gastrocnemius. (D) The vertical and horizontal right heel marker data of Participant 3.

perturbation time should occur. Since we could not know the exact stance duration while the perturbed gait cycle was occurring, we verified that perturbations were delivered at the correct time during post processing. During each perturbation, the right belt speed will increase from 1.2 m/s to 1.95 m/s for 25 ms at a rate of 15 m/s². The speed remains at 1.95 m/s for 75 ms, then decreases to 1.2 m/s for 25 ms, at a rate of 15 m/s². The left belt remains at a constant 1.2 m/s throughout the entire trial. The complete Lua script that generated the perturbations is included in Supplemental File 1. A sample of the ground reaction forces and belt speed can be seen in Fig. 3, the code to generate this figure is available in Supplemental File 2.

## Protocol
### *Experimental setup*
The experimental protocol consisted of seven 5-min trials of perturbed walking. Before each participant's data-gathering session, the motion capture system was calibrated using

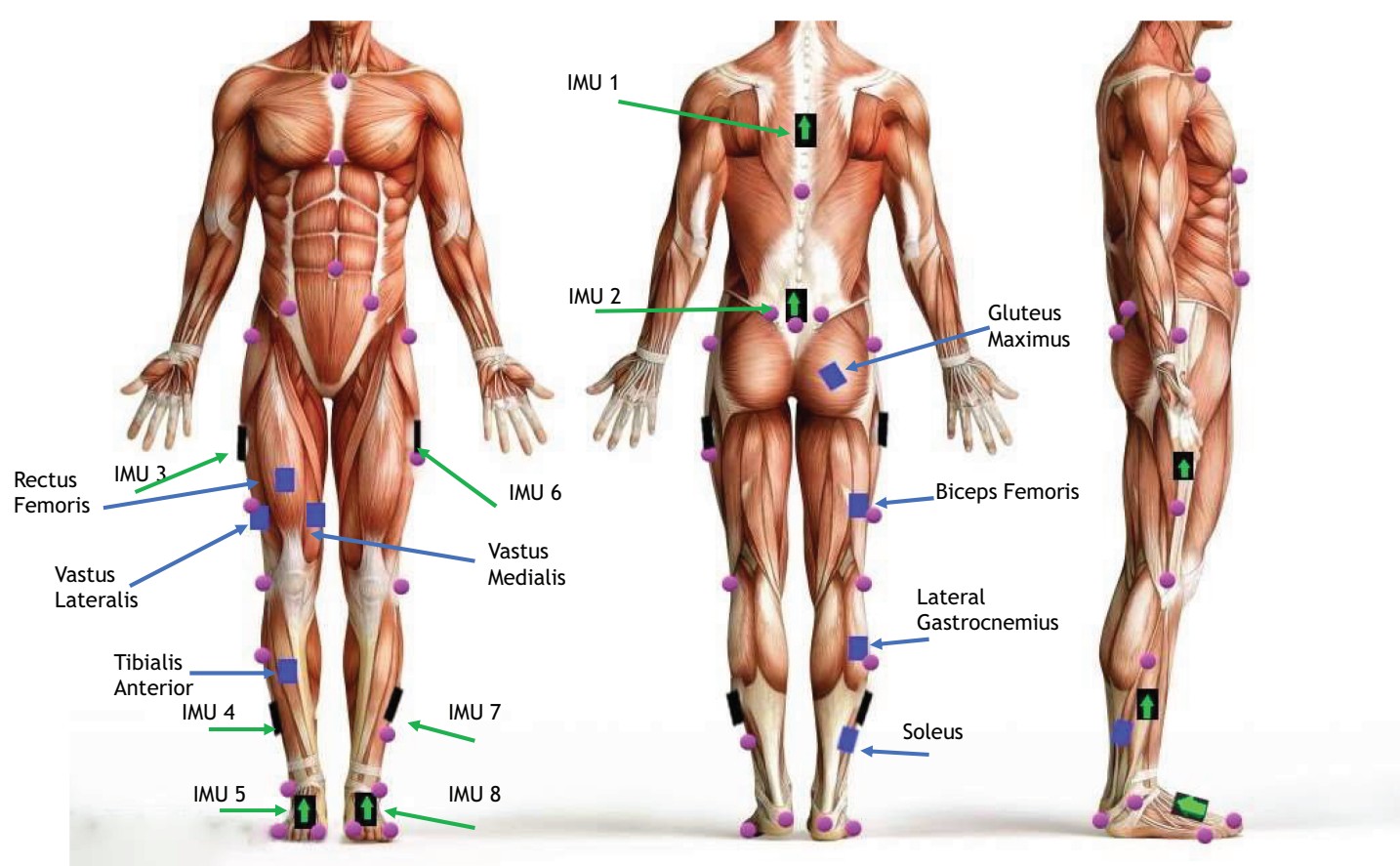

**Figure 4 Marker and sensor setup.** A depiction of marker locations (purple), EMG sensor (blue), and IMU (black with green arrow) placement on participants. Image Source Credit: Copyright iStockphoto.com/portfolio/cosmin4000. 

the manufacture's recommended procedure. Once participants arrived for their data-gathering session, they changed into athletic shoes, shorts, and a t-shirt. Before changing, participants were given an EMG and instructions on where and how to apply an EMG on their gluteus maximus muscle.

Participants were then asked to step into a full-body safety harness. After adjusting the straps to fit, they were asked to take a few steps to ensure that the harness did not affect their gait. Once in the harness, eight wireless EMG sensors were placed on the right leg. The sensors were placed on the belly of the following muscles: tibialis anterior, lateral gastrocnemius, soleus, rectus femoris, vastus lateralis, vastus medialis, biceps femoris, and gluteus maximus. Locations for these sensors/muscles can be seen in Fig. 4. Eight IMUs were placed on the participants in the following locations: on the trunk at T10, the sacrum, left and right thigh, left and right shank, and left and right foot, as seen in Fig. 4. The EMGs and IMUs were wrapped in pre-wrap to reduce the risk of sensors falling off during the trials. Twenty-five reflective markers were then placed on anatomical landmarks on the body according to the lower body marker set, as seen in Fig. 4; see Table 2 for anatomical landmark names and locations.

**Table 2 Descriptions of the names and locations of the 25-marker, marker set used in this study.**

| Marker number | Marker name | Anatomical location |
| --- | --- | --- |
| 1 | T10 | T1O |
| 2 | SACR | Sacrum bone |
| 3 | NAVE | Navel |
| 4 | XYPH | Xiphoid process |
| 5 | STRN | Sternum |
| 6 | LASIS | Pelvic bone left front |
| 7 | RASIS | Pelvic bone right front |
| 8 | LPSIS | Pelvic bone left back |
| 9 | RPSIS | Pelvic bone right back |
| 10 | LGTRO | Left greater trochanter of the femur |
| 11 | FLTHI | Left thigh |
| 12 | LLEK | Left lateral epicondyle of the knee |
| 13 | LATI | Left anterior of the tibia |
| 14 | LLM | Left lateral malleolus of the ankle |
| 15 | LHEE | Left heel |
| 16 | LTOE | Left toe |
| 17 | LMT5 | Left 5$^{th}$ metatarsal |
| 18 | RGTRO | Right greater trochanter of the femur |
| 19 | FRTHI | Right thigh |
| 20 | RLEK | Right lateral epicondyle of the knee |
| 21 | RATI | Right anterior of the tibia |
| 22 | RLM | Right lateral malleolus of the ankle |
| 23 | RHEE | Right heel |
| 24 | RTOE | Right toe |
| 25 | RMT5 | Right 5$^{th}$ metatarsal |

**Note:**
The column labeled marker name matches the column headers in the motion capture data files at Zenodo, doi: 10.5281/zenodo.10557913.

### Testing protocol

Once the instrumentation and markers were placed on the participant, the force plate was zeroed. Participants were then asked to step onto the treadmill, and the safety rope was attached to the harness so that no additional forces were acting on the participant when walking but would prevent a complete fall if their balance was lost at any point during the trials.

After securing the harness, participants were given a verbal countdown to the start of the trial. When the trial begins, the treadmill belts accelerate to 1.2 m/s at a rate of 0.2 m/s$^2$, and the recording of EMG, IMU, motion capture, force plate, and treadmill belt speed data begins. There are then 15 s of walking before the perturbations will start occurring. Perturbations occur randomly during the stance phase of the participant's right leg for the remainder of the trial, as described above. Five minutes from the start of the trial, both belts will decelerate at a rate of 0.2 m/s$^2$ until the belt speeds are 0 m/s. Participants were then

given at least a 5-min break between trials to rest and for the investigator to ensure none of the sensors became unsecured during the trial.

## RESULTS

Here, we present some basic results. A description of the raw data, along with an overview of some computed variables, is given.

### Raw data

The raw data consists of tab-delimited text files output from the 'Mocap' and 'Record' modules in D-Flow. There is one of each file for every trial. The 'Mocap' module outputs two text files, one containing the force plate and marker data and the other containing the EMG analog data. The file from the 'Record' module contains the belt speed data and information about if a perturbation occurred and when it occurred.

#### D-flow mocap module

Data were recorded as previously described in *Moore, Hnat & van den Bogert (2015)*. Specifically, the first file output from the D-flow mocap module is stored in a tab-delimited text file named Sub##_000T_Mocap.txt, where ## is the participant's identifying number and 'T' is the trial number. The file contains time series, and the numerical values with precision of six decimal places. The first line of the file contains column headers for the columns containing the corresponding data.

The TimeStamp labeled column is from the computer clock time when D-Flow receives frames from Cortex; with a sampling rate of about 100 Hz. We recommend replacing the time stamps by exact multiples of 100 Hz, which better reflects when the marker images were captured. The FrameNumber column contains the frame numbers that correspond to those from Cortex. The columns corresponding to marker coordinates are labeled with the marker name, as seen in Table 2, and are followed by. PosX, .PosY, or .PosZ, to indicate the marker's position in space. These values are given in meters and expressed in Cortex's Cartesian reference frame (Fig. 1). If a marker was blocked or dropped during the data gathering session,it is filled with zeros for the frame number(s) it was dropped.

The conversion of the voltages from the force plate load cells is done with a force/moment calibration matrix in Cortex. The columns with headers that start with FP1 or FP2 contain the ground reaction force and moment data for the left force plate (FP1) and the right force plate (FP2). The columns that contain the ground reaction force data have the label FP1/FP2 followed by .For[X], .For[Y], or .For[Z] with the values being given in Newtons. The columns that contain the moment from the force plates data have the title FP1/FP2 followed by .Mom[X], .Mom[Y], or .Mom[Z] with the values being given in Newton-meters. This file also contains center of pressure data, given in meters, these columns are labeled FP1/FP2 followed by .Cop[X], .Cop[Y], or .Cop[Z].

The second file output from the D-flow mocap module is stored in a tab-delimited text file named Sub##_000T_EMG.txt, where ## is the participant's identifying number and 'T' is the trial number. The file contains time series, and numerical values provided with three decimals of precision. The first line of the file contains the names of the muscles for each column of EMG data.

The column labeled TimeStamp is from the computer clock time when D-Flow receives frames from Cortex; at a sampling rate of 1,000 Hz. We recommend not using this time stamp, but replacing it with exact multiples of 1 ms, to better reflect when the analog data were captured. The FrameNumber column contains frame numbers that correspond to those from Cortex. EMG signals are recorded in Volts, but should be considered as arbitrary units due to unknown gains and conversion factors. It is recommended to use a bandpass filter on the raw EMG data to remove DC drift and noise. The raw EMG signal from the Delsys analog outputs is delayed by 48 ms relative to the motion capture and force plate signals. This delay can be compensated for during processing by subtracting 48 ms from the timestamps in the EMG data file.

### D-flow record module

The file output from the D-Flow record module is a tab-delimited text file named Sub##_000T_dflow.txt, where ## is the participant's identifying number and 'T' is the trial number. The file contains time series, and the numerical values with precision of six decimal places. The first line of the file contains column headers for the time stamp column, treadmill belt speed, and information from the D-Flow script module (*i.e.*, command belt speed, stance duration, state variable, and perturbation information).

The column labeled TimeStamp is from the computer clock time when D-Flow receives frames from Cortex; the sampling rate for these is approximately 300 Hz. The columns labeled LeftBeltspeed and RightBeltspeed contain the speed of the left and right treadmill belts, respectively, and are given in m/s. The CommandSpeed column contains the speed that the script module in D-Flow wants the right treadmill belt speed to be, for example when a perturbation occurs the command speed will jump from 1.2 to 1.95 m/s. This command is then executed with the maximum acceleration of 15 m/s$^2$ that the treadmill is capable of, reflected in the RightBeltspeed data. The column labeled StanceDuration is what the stance duration of the participant was calculated to be within the script module based on the history of vertical ground reaction force from the right force plate. The column labeled State is the state variable of the perturbation script, which can be seen in Supplemental File 1. The column labeled Test is a number that tells us if a perturbation occurs; a one in this column indicates that a perturbation occurred during that stance phase, 2–6 means no perturbation occurred during that stance phase. The column labeled Perturbationtype is a numerical value that corresponds to when the perturbation occurred during the perturbed gait cycles; 1–perturbation at 10% of stance phase, 2–a perturbation at 20% of stance phase, …, 8–a perturbation at 80% of stance phase.

### IMU sensor data

The IMU sensor data file output is a comma-delimited text file named Sub##_000T_IMU.txt, where ## is the participant's identifying number and 'T' is the trial number. Each trial has one text file containing data from all eight IMU sensors. The first column for the first several lines contains sensor information for each IMU channel, including the label

(ex. Trigno IM sensor 1: Acc 1.X (IM)), sampling frequency, number of points, and units. The system gain, a/d card gain, bit resolution, bias, high pass cutoff frequency, and low pass cutoff frequency can be found in the first column.

The data from the sensors that can be analyzed starts on line 564 and is in columns 1–160. Each sensor channel has two columns associated with it. The first of the two columns, labeled X[s], contains the time stamps corresponding to the channel's sampling rate. The second of the two columns is labeled with the label corresponding to the sensor channel (ex. Trigno IM sensor 1: Acc 1.X (IM)); this column contains the sensor data from that channel.

### Processed data

Basic analysis is shown in this section to show that the methods and data described can provide meaningful results. Data analysis was performed using MATLAB (Mathworks, Natick, MA, USA). The EMG signals were advanced 48 ms to compensate for the known lag in the Delsys system. The raw EMG signal was filtered using a second-order bandpass Butterworth filter (30–300 Hz), rectified, smoothed with a second-order lowpass Butterworth filter (20 Hz) to obtain the EMG envelope. All filters were run forward and backward in time to eliminate time lag.

The vertical and horizontal force data, EMG envelope, and motion capture data for each participant were divided into gait cycles using the ground reaction forces recorded from the right force plate. For each gait cycle, the angle of the trunk, right/left hip, right/left knee, and right/left ankle are calculated from the sagittal plane marker coordinates (*Flux et al., 2021*). Then, using the perturbation information recorded from D-Flow, each gait cycle was labeled based on the kind of perturbation that occurred during that gait cycle: no perturbation, 10%, 20%, … 80%. For each perturbation type, the corresponding gait cycles were averaged after removing outliers. EMG was normalized to the peak of the non-perturbed average. The standard error of the mean (SEM) was calculated for all averages. Figure 5 shows average EMG, joint angles and ground reaction forces for each perturbation type. The code to generate this figure can be found in Supplemental File 2.

Typically, for each participant, there are 1,600+ gait cycles without perturbation, and 22–30 gait cycles for each type of perturbation. The averaged belt speed signals in Fig. 5 demonstrate that the perturbations were consistently delivered at the same time, though not perfectly. Some variation in perturbation timing is inevitable because the timing is specified as a fraction of the stance phase. Stance phase duration is estimated in real time, from previous gait cycles and cannot predict the duration of the stance phase during which the perturbation is delivered.

## DISCUSSION

This dataset contains motion, ground reaction force, IMU, and EMG data from normal and perturbed gait data, where perturbations were generated at controlled times during randomly selected gait cycles. The EMG and motion data for the unperturbed gait are consistent with what has been seen in previous studies on normal gait (*Winter & Yack, 1987*; *Wang & van den Bogert, 2020*). The EMG data for the perturbed gait shows evidence

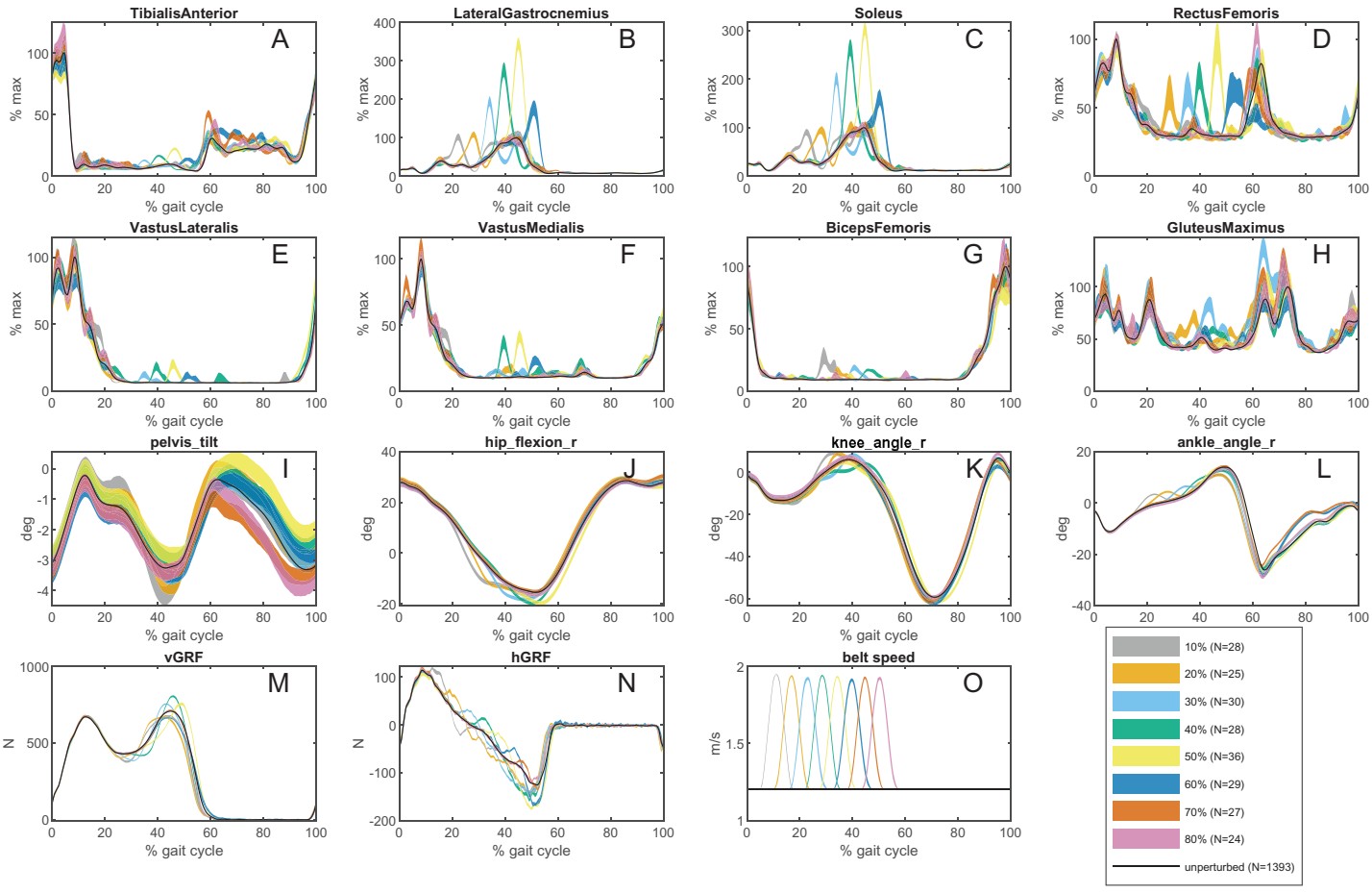

**Figure 5 An example of processed data from one participant.** (A–H) An example of the processed EMG for all observed muscles for all perturbation types for Participant 3. The black curve is the average EMG signal for the unperturbed gait cycles; each colored curve corresponds to the average ± SEM for each perturbation type. (I–K) An example of the calculated hip, knee, and ankle joint angles on the perturbed side from the motion capture data for all perturbation types for Participant 3. The black curve is the average joint angle for the unperturbed gait cycles; each colored curve corresponds to the average ± SEM for each perturbation type. (L and M) An example of the horizontal and vertical ground reaction forces on the perturbed side for all perturbation types for Participant 3. The black curve is the average ground reaction for the unperturbed gait cycles; each colored curve corresponds to the average ± SEM for each perturbation type. (O) The recorded belt speed during all perturbation types for Participant 3. The black curve is the belt speed for the unperturbed gait cycles; each colored curve corresponds to the average ± SEM for each perturbation type.

that a reflex occurs (Fig. 5), as shown for a simpler protocol in *Sloot et al. (2015)*. The number of gait cycles with the same perturbation timing (22–30) appears to be large enough to obtain a reliable average of the EMG response, despite the inherent noisiness of EMG.

The EMG data, after averaging, shows evidence that the magnitude of the reflex response is dependent on when the perturbation occurred during the stance phase (Fig. 5). This aligns with what has been seen in H-reflex studies (*Belanger & Patla, 1984*; *Zehr, Komiyama & Stein, 1997*; *Capaday & Stein, 1986*). However, in our study, mechanical perturbations were applied, which is a more ecological approach to induce and study

reflexes during gait, and how the reflex gains may be modulated during the gait cycle. It is also possible that the muscle activity that is seen in response to the perturbations could be proportional to other sensory inputs or reactions after sensory integration, and is not only due to a stretch reflex (*Safavynia & Ting, 2013*). Further analysis on the data should be done to determine if this is the case.

One appealing application of the data would be the development of human-inspired and intuitive reflex control systems for assistive devices, such as prostheses and exoskeletons (*Geyer & Herr, 2010*; *Ferris et al., 2012*). Typically, reflex control parameters are tuned or optimized to perform predictive simulations of human gait, using objectives such as minimal metabolic energy expenditure (*Geyer & Herr, 2010*; *Veerkamp et al., 2021*). Even when the simulated gait looks good, the controller may not perform well during perturbations, in simulation or in robotic devices. We suggest that feedback controllers for robotic assistive devices can be optimized to generate movement responses after each perturbation that matches the human responses in this dataset.

The provided dataset contains IMU data, which we have not performed any analysis on, but could be useful for evaluating methods for processing IMU data. IMU technology is cheap and increasingly used in clinical motion capture applications. Our data may help validate the use of IMU data for perturbation responses. IMU processing typically uses machine learning methods, trained and validated on normal gait data (*Cho et al., 2018*), which can introduce bias towards normal gait in clinical applications. This dataset can test the capability of ML to correctly quantify unusual or unexpected features in human gait.

Some technical limitations should be taken into consideration. First, it should be noted that the GRF data, specifically the anterior-posterior COP, is unreliable during belt acceleration and deceleration (*Hnat & van den Bogert, 2014*). Second, there is not IMU data for all participants. This is from an issue that arose with the IMU sensors dying during the data gathering sessions, leading to incomplete trial sets for participants 2, 15, 90, and 99. Finally, due to the large volume of data, careful curation is needed. Markers were occasionally dropped. Sometimes, a participant might touch the force plate during the swing phase, or take an abnormal step, which then results in an incorrectly timed perturbation. In the code that generated Fig. 5 (Supplemental File 2), such situations were automatically detected and excluded from the gait cycle averaging process.

## CONCLUSION

We have presented a comprehensive dataset of motion, ground reaction loads, and EMG from human participants during perturbed walking. The raw data from this study is available publicly for reuse. Although this data has multiple other uses, we believe it is ideal for identifying control laws. Researchers working on mathematical models for gait control may find this data useful as a way to generate and validate their models.

## ACKNOWLEDGEMENTS

We want to thank Rex Boyer for his assistance with data gathering.

### Funding

The work was supported by a Faculty Research Development grant from Cleveland State University. There was no additional external funding received for this study. The funders had no role in study design, data collection and analysis, decision to publish, or preparation of the manuscript.

### Grant Disclosures

The following grant information was disclosed by the authors:
Faculty Research Development grant from Cleveland State University.

### Competing Interests

Antonie J. van den Bogert is an Academic Editor for PeerJ.

### Author Contributions

- Dana L. Lorenz conceived and designed the experiments, performed the experiments, analyzed the data, prepared figures and/or tables, authored or reviewed drafts of the article, and approved the final draft.
- Antonie J. van den Bogert conceived and designed the experiments, analyzed the data, authored or reviewed drafts of the article, and approved the final draft.

### Human Ethics

The following information was supplied relating to ethical approvals (*i.e.*, approving body and any reference numbers):

The study was approved by the Institutional Review Board of Cleveland State University (IRB-FY2019-178).

### Data Availability

The data is available at Zenodo: Lorenz, D. L., & van den Bogert, A. J. (2023). A Comprehensive Dataset on Biomechanics and Motor Control During Human Walking with Discrete Mechanical Perturbations [Data set]. Zenodo. https://doi.org/10.5281/zenodo.10557913.

### Supplemental Information

Supplemental information for this article can be found online at http://dx.doi.org/10.7717/peerj.17256#supplemental-information.

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
