# Peer review of "A comprehensive dataset on biomechanics and motor control during human walking with discrete mechanical perturbations"

_PeerJ, doi:10.7717/peerj.17256_

## Round 0.1 · original submission · Minor Revisions

Both reviewers have asked you to provide a more comprehensive description of perturbation responses during walking beyond "local reflexes" and to provide references to comparable databases and have made additional suggestions for further improvement of the paper.

Reviewer 1 ·

Basic reporting

clear

Experimental design

clear

Validity of the findings

N/A

Additional comments

The authors of this paper provide an comprehensive dataset on the whole-body response to sudden changes in treadmill belt speed during walking .The authors share this data in a very clear and structured way. I have several remarks that might improve the quality of the manuscript.

Major remarks:
• The authors wrote that mechanical perturbations is a more ecological approach to study reflexes during functional activities (L 45). While citing Sloot et al., who associated rapid changes in treadmill belt speed with inducing reflexes due to changes in muscle fibre kinematics, an alternative explanation suggests that muscle activity in response to perturbation may be proportional to other sensory inputs (e.g., the vestibular system) or reactions after sensory integration (e.g task level variables) ( mainly studied in standing balance, for example in [1] ). Addressing these potential mechanisms in the introduction or discussion section could enhance the manuscript by broadening the understanding behind observed changes in muscle activity after perturbation.
• The authors do not mention the size of the treadmill belts in the experimental setup. Although the treadmill appears sufficiently sized in figure 1, clarifying its dimensions could be important as the treadmill size might influence the subjects response to the perturbation. Including the treadmill dimensions in the experimental setup section would be beneficial.
• Please add a label to the horizontal axis in figure 3

Minor suggestions: (feel free to ignore these more personal comments/suggestions)
• Expand the references to studies employing similar perturbations on healthy subjects during various moments in the gait cycle. A more extensive list of studies could aid readers in pooling data from multiple sources.
• Consider improving the column names in the _analog.txt files to establish a more direct correlation between the analog channel number (header in the txt file) and the respective location of the EMG sensor. (I realize that this relation is written in the manuscript but this is not very convenient)
• The _analog txt file also contains several columns without data ( I think Channel 48-77 if I understand the column labelling correctly). You could consider removing these columns as it only increases the size of the whole dataset.
• (Note that the two points above also apply for the _mocap.txt files)
• You will probably add a short readme to the zenodo repository with a reference to the publication (as soon this is published).
• Proper headers for the IMU files (instead of sensor numbers) would it also make it a lot easier for everyone that will use the data.

References:
[1] S. A. Safavynia and L. H. Ting, ‘Long-latency muscle activity reflects continuous, delayed sensorimotor feedback of task-level and not joint-level error’, Journal of Neurophysiology, vol. 110, no. 6, pp. 1278–1290, 2013, doi: 10.1152/jn.00609.2012.

Reviewer 2 ·

Basic reporting

This article presents a dataset containing biomechanical and electromyographic measurements from 10 young adults who responded to perturbations while walking on a treadmill. The article is generally written well, but there are some basic reporting issues that should be addressed.

First, it is unclear in the introduction why the authors focus on spinal reflexes when responses to perturbations during walking are far more complex and require multiple steps. Given that the data being shared contain kinematics and kinetics, the authors should reorganize the introduction to more clearly explain the need for studying biomechanical and neurophysiological responses to perturbations during walking.

Second, please expand the paragraph in the introduction describing prior published perturbation data to acknowledge other openly available datasets of perturbed walking.

Leestma JK, Golyski PR, Smith CR, Sawicki GS, Young AJ. Linking whole-body angular momentum and step placement during perturbed human walking. J Exp Biol. 2023 Mar 15;226(6):jeb244760. doi: 10.1242/jeb.244760. Epub 2023 Mar 29. PMID: 36752161; PMCID: PMC10112983.

Liu C, Park S, Finley J. The choice of reference point for computing sagittal plane angular momentum affects inferences about dynamic balance. PeerJ. 2022 May 12;10:e13371. doi: 10.7717/peerj.13371. PMID: 35582618; PMCID: PMC9107787.

Experimental design

Because this manuscript is submitted as a dataset alone, the methods are not described as being hypothesis testing.

Validity of the findings

Because this manuscript is only meant to describe the dataset, there are no conventional results to speak of.

Additional comments

Abstract
Line 17: Please change “research data” to “publicly available research data.”

Introduction
Lines 36 – 37: Add citations to support this statement
Line 39: Quantifying reflexes is not particularly challenging. The specific challenge is to quantify them during dynamic tasks. Please clarify this point in the text.
Line 53: Add “throughout the gait cycle” after “modulated” and add “from prior studies” after “available.”
Line 62: Change “none of these provide” to “there are limited publicly available data.”
Line 68: Please use a quantitative measure of perturbation size and be sure to describe the size of the perturbations in the current study.
Lines 73 – 77: The authors should clarify what they mean by "enable development of clinical applications" and “neural control identification,” as the meaning of these phrases may not be easily understood by the reader.

Methods
Please state the rate at which each of the signals was sampled.
Line 88 – 95: Please explain what the ranges reported in this paragraph correspond to.
Line 126: How was the actual phase of the perturbation verified? Given that the gait phase has to be estimated in real-time, there is guaranteed to be some error between the commanded and actual phase of the perturbation.

Results
Lines 196 – 198: Are these data of use to potential end users? If not, they should be deleted and if so, please state precisely what was recorded on each channel.

Discussion
Lines 279 – 282: These sentences refer to two very different applications of reflex controllers: the first for predictive simulations and the second for control of assistive devices. As a result, this paragraph should be reworked to explain how reflex controllers are developed for assistive devices and how the data provided by this study might influence this process.

Figure 3: Please increase the font size, change the x-axis to have units of seconds, and add panel labels.
Figure 5: This figure is impossible to read. Please move the kinematics, and GRF data to a separate plot and make all of the fonts much larger so that the text is legible. Please also explain what the colors correspond to in the caption.

---

## Round 0.2 · Minor Revisions

One of the reviewers has given very specific comments regarding the emphasis on reflexes and I agree with the reviewer that amendments are needed here. Balance recovery responses include reflexes, but also include slower and more complex responses, based on more than local feedback related to joint angle changes, in fact these appear to more important for balance recovery.

Reviewer 1 ·

Basic reporting

no comment

Experimental design

no comment

Validity of the findings

no comment

Additional comments

The authors have addressed my concerns in their revised manuscript

Reviewer 2 ·

Basic reporting

The authors have addressed most of my primary concerns from the initial submission. However, they still largely motivate their study with the idea of examining reflexes despite the fact that the data set provides more much extensive information.

The authors should change the first sentence of the abstract to highlight the role of reactive responses (not just reflexes) and the need for biomechanical data capable of characterizing these reactive responses. Similarly, the first three paragraphs of the introduction still focus almost exclusively on reflexes. To place the paper in a broader context, I suggest removing the second paragraph and replacing it with a description of why reactive control strategies beyond reflexes are important in balance control. Then, the paragraph beginning with “a more ecological approach” could focus on reactive responses more broadly and not just reflexes.

Experimental design

No comment

Validity of the findings

No comment

---

## Round 0.3 · accepted · Accept

Thank you for addressing the remaining comments of the reviewer. This paper and for sharing the associated data set. I believe that this will make a valuable contribution.